# Study of the Durability Damage of Ultrahigh Toughness Fiber Concrete Based on Grayscale Prediction and the Weibull Model

Chen Wang [1], Pei Fu [1,*], Zeli Liu [1], Ziling Xu [1], Tao Wen [2], Yingying Zhu [3], Yuhua Long [1] and Jiuhong Jiang [1]

[1] School of Civil Engineering, Architecture and Environment, Hubei University of Technology, Wuhan 430068, China; 102010862@hbut.edu.cn (C.W.); 101910585@hbut.edu.cn (Z.L.); 101910580@hbut.edu.cn (Z.X.); 101900545@hbut.edu.cn (Y.L.); 19910019@hbut.edu.cn (J.J.)

[2] China Construction Shenzhen Decoration Co. Ltd., Wuhan 430068, China; a15102421101@163.com

[3] Wuhan Municipal Road & Bridge Co. Ltd., Wuhan 430068, China; zy601520162@163.com

* Correspondence: 20050012@hbut.edu.cn

**Abstract:** The purpose of this research is to investigate the durability damage law for ultrahigh toughness cementitious composites (UHTCCs) under freeze–thaw environments and impact resistance. In this study, UHTCCs with fiber length-to-diameter ratios of 5/30, 8/30, 12/20, 12/30 and 12/48 were tested for impact resistance and freeze–thaw cycles. The freeze–thaw cycle process and impact resistance process for UHTCC are comprehensively analyzed and evaluated in terms of mass loss, compressive strength loss, relative dynamic elastic modulus loss and impact resistance number. The freeze–thaw damage prediction model for the relative dynamic elastic modulus of the UHTCC is established based on the regularity of the measured data for the relative dynamic elastic modulus of UHTCC and also on the GM(1,1) power model. The accuracy and reliability of the GM(1,1) power model is analyzed using the relative error, absolute correlation degree, mean variance and probability of small errors. According to the evolution law of the impact resistance number of the UHTCC, the impact damage prediction model for UHTCC is established based on the Weibull distribution model, and the accuracy of the model is analyzed by using the decision coefficient R[2]. The results show that UHTCC has high durability performance, and the durability performance of UHTCC at a length-diameter ratio of 12/48 is optimal. The freeze–thaw damage evolution model and impact damage evolution model established in this research are sufficiently realistic, the average relative error of the GM(1,1) power model is less than 5%, and the coefficient of determination $R^2$ of the Weibull distribution model is greater than 0.93, which effectively reflects the damage development process for concrete under freeze–thaw and impact environment with high fitting accuracy.

**Keywords:** ultrahigh tenacity fiber concrete; durability; freeze–thaw damage; impact resistance; GM(1,1) power model; Weibull distribution model





## 1. Introduction

Since the 21st century, the use of concrete materials in the construction industry with their own characteristics of high compressive strength and plasticity has seen increasing demand. However, the internal porous multiphase structure of concrete can lead to a short service life with poor durability performance of the short board [1–3], which undoubtedly greatly restricts the application of concrete in practical engineering. When the durability performance is insufficient, it is very likely that structural damage will occur under normal loading, so the durability of concrete performance puts forward higher requirements. To solve the characteristics of concrete brittle damage and ease of cracking, the crack extension rate is delayed under the action of freeze–thaw cycles and impact, and the crack width is controlled so that the concrete brittle damage becomes ductile damage. Therefore, the new engineered cementitious composite (ECC [4]) proposed by Professor Victor C. Li from the University of Michigan in the United States leads to ultrahigh toughness fiber-reinforced concrete (UHTCC). Many concrete material research scholars have found that this material

has high toughness, high elasticity, high crack resistance, high durability and other excellent characteristics. Under the action of shear loading, it has a very strong resistance to crack development and strain hardening phenomena and can greatly improve the brittleness of concrete and improve the ductility of concrete [5–9].

Currently, scholars in the field of concrete materials mainly study the mechanical and mechanical properties of materials based on UHTCC and investigate how to improve the strength of UHTCC in terms of the matching ratio [10], fiber type [11] and mixing method [12]. Suthiwarapirak et al. [13] prepared specimens by spraying to understand the fatigue fracture performance of ECC and polymer cement mortars and measured the fatigue damage mechanism, damage extension and fatigue life. The structures showed that ECC has excellent fatigue resistance and its fatigue failure behavior is similar to that of metals. Jaehyun Lee et al. [14] investigated the compressive strength properties of binary low carbonate concrete substituted with blast furnace slag (GGBS) and fly ash (FA), which is good for reducing $CO_2$ emissions and construction costs, investigated the compressive strength properties of binary mixed low carbonate concrete based on the substitution ratio, and derived the applicable mixing ratio range. Babar Ali et al. [15] created concrete composites (FCCs) by adding 0.5 and 1.0% volume fraction of glass fibers (GF), hooked steel fibers (HSF) and polypropylene fibers (PPF) to normal strength concrete (C30); concrete with GF-FCC and PPF-FCC is more environmentally friendly than conventional concrete for the same load-bearing capacity and more economical. Iman Ferdosian, Aires Camões [16] showed the mechanical properties of an eco-efficient combination of self-compacting ultra-high toughness fiber concrete made from ultrafine fly ash and low cement content using a 4-point flexure test to evaluate the first cracking strength, toughness indices and residual strength factors, and the results showed that the concrete has a very high energy absorption capacity, tensile strength and residual flexural tensile strength. The mechanical performance of the concrete was greatly improved. Bhanavath Sagar, M.V.N. Sivakumar [17] analytically evaluated the variation patterns of workability, compressive strength, flexural strength, split tensile strength, load deflection and uniaxial stress–strain curves for PVA-FRC. PVA-FRC concrete with 0.3% PVA fiber content showed good mechanical performance and it was possible to more accurately predict the relationship between the material parameters and improved reinforcement index. At this stage, the focus of research on UHTCC is still on its mechanical performance, and the strength indices and ultimate stress–strain have indeed been greatly improved, but research into its durability has been neglected. Freeze-thaw cycles, impact damage, acidic environments and carbonation have a great impact on the durability of concrete structures.

Gray system theory focuses on the study of small samples, poor data, and uncertainty. It is characterized by small data modeling, based on partially known data, through the role of sequence operators to explore the realistic laws of things in motion. Engineering applications overcome the problems of small samples and incomplete information and have good feasibility and accuracy. The GM(1,1) power model is a mathematical prediction model based on gray parameters [18], gray equations [19] and gray matrices [20]. The GM(1,1) power model is also the most widely used prediction model in gray system theory, on which scholars often base their damage models for damage life prediction. Yushi Yin et al. [21], in an experiment to study the effect of sulfate on the mechanical properties of concrete, introduced a GM(1,1) power model to establish a concrete damage prediction model based on gray system theory. The results showed that the compressive strength loss of C80 concrete is decreased by 27.4% and 30% within 360 and 720 wet and dry cycles, respectively, the mechanical properties of concrete are greatly improved, and the late deterioration of high strength concrete is very slow. Mingxi Liu et al. [22] used the image segmentation algorithm of the square area to derive the voids for porous asphalt concrete and used the gray entropy method to analyze the effect of different equivalent diameter voids on the sound absorption performance of porous asphalt concrete in the range of traffic noise. The results showed that the average sound absorption coefficient of porous asphalt concrete increases with increasing air, the sound absorption performance

is improved, and the sound absorption performance is mainly affected by the equivalent diameter of 3–4 mm. Baoyang Yu et al. [23] studied the frost behavior of permeable asphalt concrete by freeze–thaw cycle tests and the water stability problems caused by spalling and loosening effects and introduced the GM(1,1) power model to evaluate the water stability of permeable asphalt concrete in seasonal freezing areas. The results concluded that permeable asphalt concrete has the best water stability with a porosity of 19–21% and the largest asphalt peel area with a porosity greater than 24%. Yan Tan, Ziling Xu et al. [24] studied the effect of silica fume and polyvinyl alcohol fiber on the mechanical properties and frost resistance of concrete. The best frost resistance for concrete was achieved with 10% silica fume and 1% fiber, the compressive strength increased was by 26.6% and the flexural strength was increased by 29.17% under the influence of the compound action. The GM(1,1) power model introduced reflects well the damage progression for concrete under the action of freeze–thaw cycles. Through reading a large amount of literature, we found that the GM(1,1) power model is very suitable for handling freeze–thaw cycling tests with equal time spacing, and the freeze–thaw damage model based on the GM(1,1) power model has high prediction accuracy and a good fit.

The Weibull distribution model was proposed by Swedish physicist Waloddi Weibull in 1939 as a theoretical basis for reliability analysis and life prediction and is widely used in reliability engineering research [25–27]. The Weibull probability density function [28] can be derived from the relationship between strength prediction and life prediction, and the reliability life prediction of the structure is carried out with high prediction accuracy. Byung Wan Jo et al. [29] used nanosilica and hydrated alumina combined with the sol-gel method to synthesize nanocement and introduced a Weibull distribution model to optimize the curing time, analyzed the variation pattern of the compressive strength of nanobased concrete and derived the prediction equation using the relationship between the compressive strength and rate of change of the curing time. It was concluded that the curing time required to achieve full strength for nanocement-based concrete is 21 d, while conventional Portland cement requires 28 d. The use of nanocement as the primary binding material can significantly reduce the time required for construction. G. Murali et al. [30] used short crimped fibers and long hooked end steel fibers in two-stage concrete (TSC) and tested them under drop hammer reloading and used the Weibull distribution to analyze the results of the dispersion tests, which showed that the use of a higher content of fibers achieved better impact resistance and good linear correlation for the test results. H.-K. Man, J.G.M.van Mier [31] used an extended lattice model to analyze the effect of size on the strength of concrete prismatic specimens subjected to 3-point bending. The skeletal structure was obtained from CT scans of concrete time, and the size effect was simulated using the Weibull model, from which the main Weibull parameters were obtained. The crack size distribution was calculated, which helped to analyze the fracture damage process in depth. An alternative macroscopic model, called the 4-stage fracture model, was proposed based on the Weibull distribution model. L.E. Zapata-Ordúz et al. [32] analyzed the effect of factors containing fly ash and silica compounds on the splitting tensile strength using the compound hypothesis to study the accuracy of the Weibull model and concluded that the average Weibull modulus does not vary significantly with time.

From the above literature, it can be seen that most of the literature mainly focuses on the mechanical properties of UHTCC, with less research on its durability. In this paper, based on the freeze–thaw cycle test and impact test, the GM(1,1) dynamic model and Weibull distribution model are introduced to establish the durability damage model and predict the durability damage characteristics under the action of a freeze–thaw cycle test and impact test, respectively.

## 2. Materials and Methods

### 2.1. Mix Proportion Design

The materials used in the preparation of UHTCC include cement, natural gravel, fine sand, modified polypropylene fibers (PP), water and solid polycarboxylic acid water-

reducing agents. The cement used is P•O 42.5R grade cement produced by the Wuhan Huaxin Cement Plant, with a standard consistency of 25% water consumption, a specific surface area greater than 300 m$^2$/kg, 3 d and 28 d compressive strengths of 25.6 MPa and 48.1 MPa, respectively. The Fly ash used the College's special first-class fly ash, a 45 μm sieve margin (%) for fineness indicators of not more than 12%, a water demand ratio of not more than 95%, and a specific surface area greater than 400 m$^2$/kg. River sand was used as a fine aggregate, with a fineness modulus of 1.85; the coarse aggregate utilized natural gravel with a good particle gradation, and the particle size of the gravel was controlled between 5 and 15 mm; the water reducing agent used was a white powdered polycarboxylic acid water reducing agent with a water reduction rate of 25%. For the fiber, a new domestic modified PP fiber with surface roughening and a Y-interface was used. The performance index of the fiber is shown in Table 1, and the concrete mix proportion is shown in Table 2.

**Table 1.** Performance indices of modified PP fiber.

| Length (mm) | Diameter (μm) | Tensile Strength (MPa) | Dynamic Elastic Modulus (GPa) | Elongation (%) | Density (g/cm$^3$) | Melting Point (°C) | Resistivity (Ω·cm) | Thermal Conductivity |
|---|---|---|---|---|---|---|---|---|
| 5, 8, 12 | 20, 30, 48 | 500 | 3.5 | 20 | 0.91 | 165–173 °C | $7 \times 10^{19}$ | Worse |

**Table 2.** Mix ratio of UHTCC.

| Type | Fiber Content (%) | Fiber Length (mm) | Fiber Diameter (μm) | Dosage (kg/m$^3$) | | | | | |
|---|---|---|---|---|---|---|---|---|---|
| | | | | Water | Cement | Fly Ash | Coarse Aggregate | Fine Aggregate | Water-Reducing Agent |
| UHTCC 5-30-1 | 1.0% | 5 | Φ30 | 265 | 450 | 495 | 425 | 950 | 10 |
| UHTCC 5-30-2 | 2.0% | 5 | Φ30 | 265 | 450 | 495 | 425 | 950 | 10 |
| UHTCC 8-30-1 | 1.0% | 8 | Φ30 | 265 | 450 | 495 | 425 | 950 | 10 |
| UHTCC 8-30-2 | 2.0% | 8 | Φ30 | 265 | 450 | 495 | 425 | 950 | 10 |
| UHTCC 12-20-1 | 1.0% | 12 | Φ20 | 265 | 450 | 495 | 425 | 950 | 10 |
| UHTCC 12-20-2 | 2.0% | 12 | Φ20 | 265 | 450 | 495 | 425 | 950 | 10 |
| UHTCC 12-30-1 | 1.0% | 12 | Φ30 | 265 | 450 | 495 | 425 | 950 | 10 |
| UHTCC 12-30-2 | 2.0% | 12 | Φ30 | 265 | 450 | 495 | 425 | 950 | 10 |
| UHTCC 12-48-1 | 1.0% | 12 | Φ48 | 265 | 450 | 495 | 425 | 950 | 10 |
| UHTCC 12-48-2 | 2.0% | 12 | Φ48 | 265 | 450 | 495 | 425 | 950 | 10 |

The modified PP fibers used in this research have a Y-shaped cross-section. The surface of the fibers was roughened without changing the tensile properties and toughness of the fibers, thus improving the bonding ability of PP fibers with UHTCC. SEM electron microscopy scans of the modified PP fiber and the normal PP fiber are shown in Figure 1. The roughness of the treated PP fiber surface increased significantly, allowing for better bonding with ultrahigh toughness cementitious composites. The fibers are distributed in a disordered manner after mixing, where the fibers of the Y-shaped cross-section fit better with the ultrahigh toughness cementitious composites. Yu J H et al. [33] showed experimentally that the most fundamental property of fiber-reinforced cementitious materials is fiber bridging matrix cracking, which represents the average force acting on the crack opening by the fibers inside the composite when subjected to tension. The contact angle of untreated PP fibers with water is 112.5°, and the contact angle of acid-treated modified PP fibers is 78.1°. That is, the better the wettability of the PP fiber after the roughening treatment, the better it can combine with the UHTCC material to have better crack resistance.

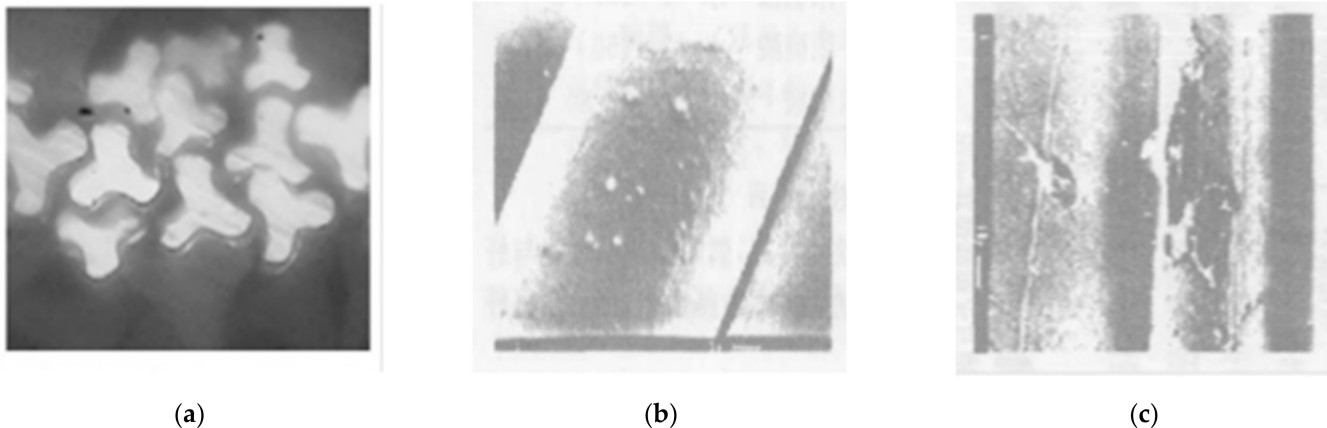

(a)  (b)  (c)

**Figure 1.** SEM comparison for PP fibers before and after surface treatment: (**a**) Y-shaped section; (**b**) ordinary PP fiber; (**c**) modified PP fiber.

### 2.2. Experimental Methods

The freeze–thaw test method used the rapid freeze–thaw method in the standard for test methods of long-term performance and durability of ordinary concrete (GB/T50082-2009) [34]. A 100 mm × 100 mm × 100 mm cubic specimen and 100 mm × 100 mm × 400 mm prismatic specimen was used, with each group of specimens maintained in a standard constant temperature maintenance room for 28 d. After maintenance, the specimens were immersed in water at (20 ± 5 °C), and the initial mass and initial dynamic elastic modulus of each group of specimens was measured before the test began. The freeze–thaw cycle test was conducted using the model TDR-III rapid freeze–thaw equipment, as shown in Figure 2a. The frost damage resistance model for UHTCC under different numbers of freeze–thaw times (0, 25, 50, 75, 100, 125, 150 times) was analyzed, and the dynamic elastic modulus, compressive strength and quality of UHTCC specimens after freeze–thaw cycling were tested. The dynamic modulus of the elasticity test is shown in Figure 2b.

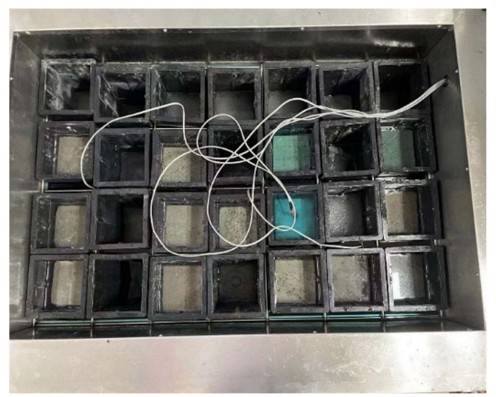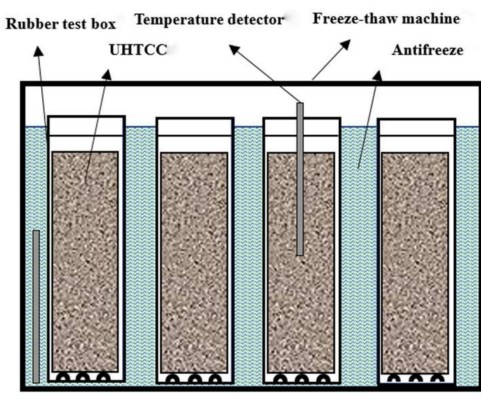

(a)

**Figure 2.** *Cont.*

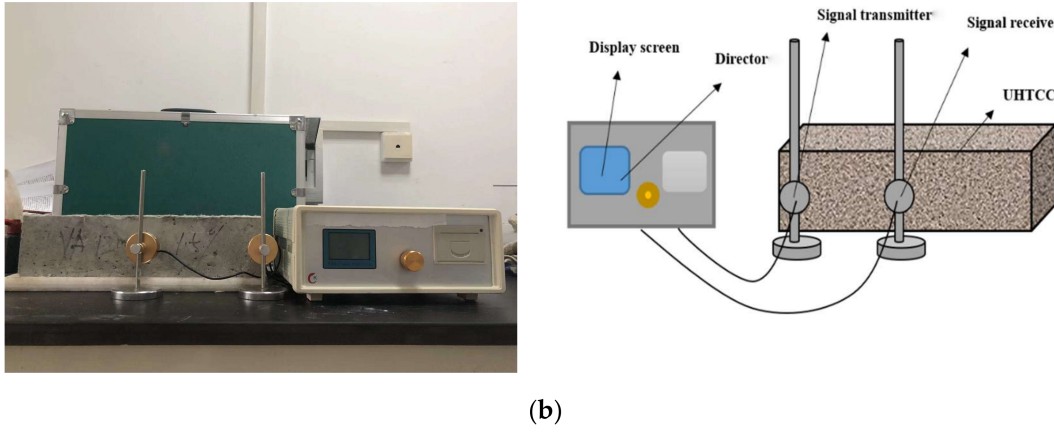

(**b**)

**Figure 2.** Schematic diagram of the test equipment: (**a**) freeze–thaw cycle test; (**b**) dynamic elastic modulus tester.

The impact test method mainly refers to the standard drop hammer impact test method recommended by the American Concrete Institute (ACI544). The method has the advantages of simple operation and relatively low requirements for the test conditions. This research utilized a CECS13-2009 concrete falling hammer impact testing machine to conduct impact resistance tests on UHTCCs, as shown in Figure 3a. The specific test steps are as follows: remove the UHTCC impact specimens from the standard curing chamber before the test, wipe the specimens and clean the fine particles present on the surface of the specimens. The CECS13-2009 concrete falling hammer impact tester was fixed, and a Φ150 mm × 63 mm cylindrical specimen was placed into the center of the chassis located inside the four single plates. The specimen was placed 5 mm away from the baffle plate, and the steel ball was placed accurately at the positive center of the specimen. The infrared ray device was turned on using the magnetic switch on the instrument so that the infrared ray device is precisely directed at the apex of the steel ball. Next, the falling position of the hammer was fixed at the specification of 500 mm. The magnetic switch controlling the falling hammer was connected to a counter, and for each press of the button, the counter noted the corresponding number of impacts and a reading was taken at the end of the test. The initial cracking state of the concrete was taken as that for when the first crack in the specimen was observed, and the number of impacts at this time was recorded. The final state of concrete cracking was taken as that when the concrete specimen was damaged to the extent that it could touch any three of the four baffles, as shown in Figure 3b, and the number of impacts at this time was recorded, together with the number of impact resistances for the final destruction of the specimen. The test finally reflects the impact resistance of concrete based on the impact energy $W$ and ductility ratio of UHTCC, and the impact energy $W$ is shown in Equation (1), and the ductility ratio is shown in Equation (2).

$$W = N_2 mgh \tag{1}$$

where $W$ is the energy consumed by impact (J).

$N_2$—Number of impacts at final cracking of the specimen (T)
$m$—Drop hammer quality (kg) is 4.5 kg
$g$—Earth's gravitational acceleration (m/s$^2$) is 9.8
$h$—Drop height of hammer (m) is 0.5 m

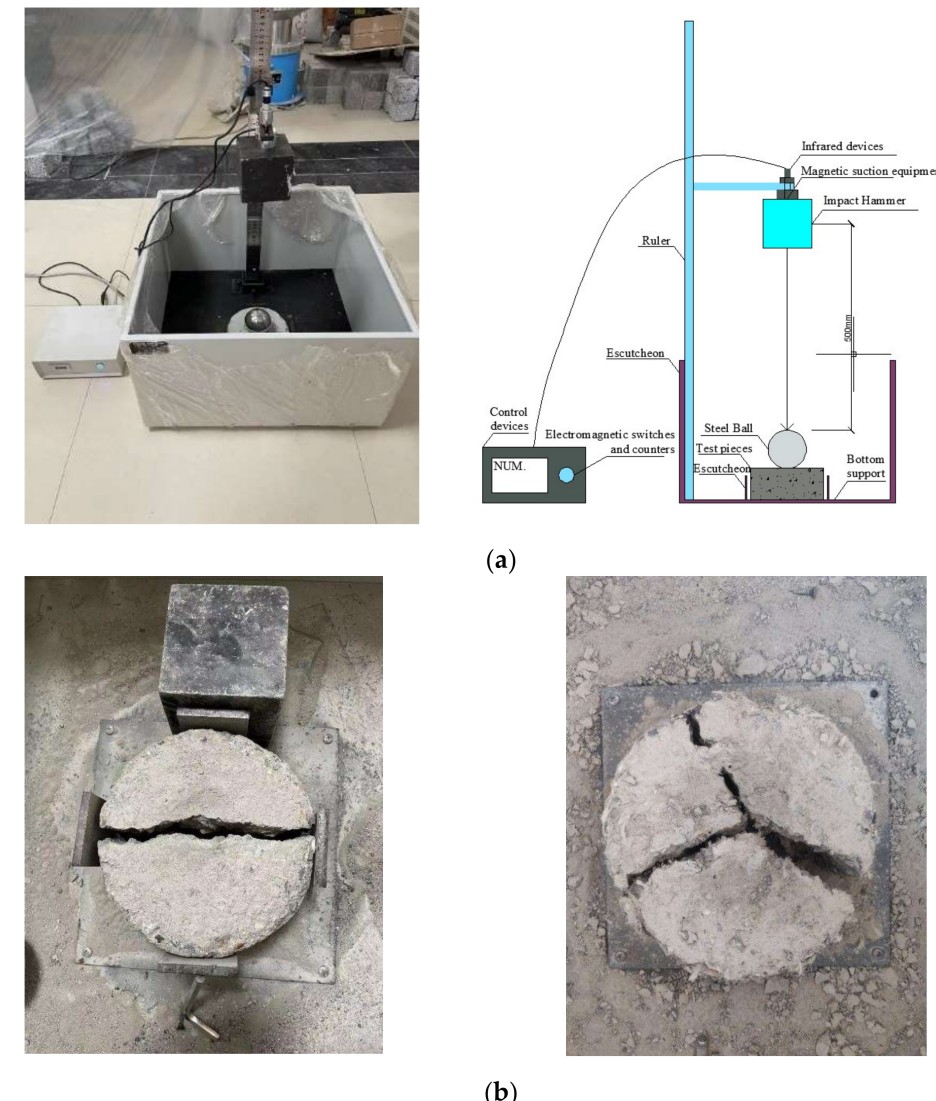

**Figure 3.** (**a**) Schematic diagram of the UHTCC impact test. (**b**) Impact test damage pattern.

$$\beta = \frac{N_2 - N_1}{N_1} \tag{2}$$

where: $\beta$ is the ductility ratio;

$N_2$—Number of impacts at final cracking of the specimen (T)
$N_1$—Number of impacts at the first crack of the specimen (T)

### 3. Test Results and Analysis

#### 3.1. Freeze-Thaw Cycle Test Results

The compressive strength, mass and relative dynamic modulus of elasticity for each UHTCC specimen after the freeze–thaw cycle test are shown in Table 3. Next, by using the number of freeze–thaw cycles as the independent variable and the mass loss, and compressive strength loss and relative dynamic elastic modulus loss as the dependent variables, a quadratic polynomial freeze–thaw damage model for UHTCC was established. The freeze–thaw damage mechanism for UHTCC is considered low cycle fatigue damage. The more representative freeze–thaw damage mechanisms are hydrostatic pressure theory, infiltration theory, water dissociation stratification theory, pore structure theory, critical saturation water value theory and water filling factor theory. The macroscopic changes

in the appearance of the UHTCC are based on the accumulation of its internal damage. Modified PP fiber incorporated into UHTCC can be used as an inducer to reduce the fluidity of the mix, convert the unfavorable large pores inside UHTCC into small pores, reduce the freezing and swelling pressure generated by the large pores during the freeze–thaw cycle and increase the compactness and frost resistance of concrete.

### 3.1.1. Freeze–Thaw Compressive Strength Damage Model

The UHTCC freeze–thaw compressive strength damage model based on quadratic polynomials under the action of freeze–thaw cycles is shown in Figure 4. The fitted decision coefficients are all higher than 0.9, which indicates a high fitting accuracy. The compressive strength damage of UHTCC specimens is relatively gentle, and there is no obvious trend of increasing compressive strength damage, which has high stability. When the number of freezing and thawing cycles reaches 100 times, cracks are formed in the internal pores of the specimen, and these cracks become channels for the free movement of pore water. When the number of freeze-thaws reaches 150, the compressive strength damage to UHTCC 8-30-1 is the most serious, reaching 50.79%, and the compressive strength damage for both UHTCC 12-30-2 and UHTCC 12-48-2 specimens shows better performance, with values of 59.26% and 62.39%, respectively. It can be observed from the test data that when the modified PP fiber length is 12 mm, the fiber diameter is 48 μm, and the fiber dose is 2%, the specimen shows better frost resistance performance. The frost resistance for UHTCC specimens with 2% fiber doping is generally better than that for UHTCC specimens with 1% fiber doping.

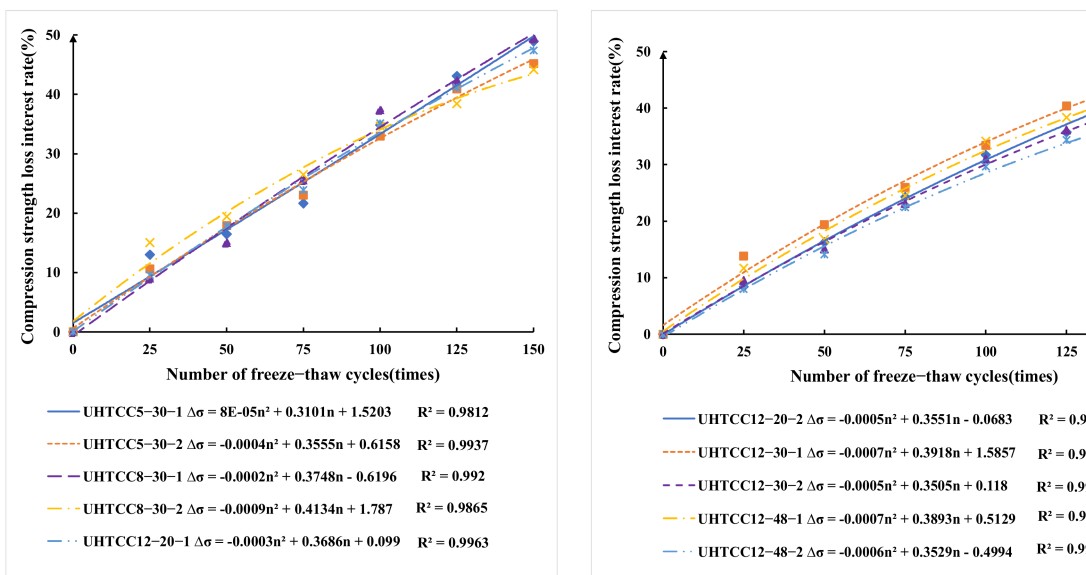

**Figure 4.** Compressive strength damage under different number of freeze-thaw cycles.

Table 3. UHTCC freeze–thaw cycle test results.

| Freeze-Thaw Cycles (n) | UHTCC 5-30-1 | | | UHTCC 5-30-2 | | | UHTCC 8-30-1 | | | UHTCC 8-30-2 | | | UHTCC 12-20-1 | | |
|---|---|---|---|---|---|---|---|---|---|---|---|---|---|---|---|
| | Compressive Strength (Mpa) | Quality (kg) | Relative Bullet (%) | Compressive Strength (Mpa) | Quality (kg) | Relative Bullet (%) | Compressive Strength (Mpa) | Quality (kg) | Relative Bullet (%) | Compressive Strength (Mpa) | Quality (kg) | Relative Bullet (%) | Compressive Strength (Mpa) | Quality (kg) | Relative Bullet (%) |
| 0 | 46.2 | 7.800 | 100 | 49.1 | 7.950 | 100 | 46.5 | 7.600 | 100 | 45.3 | 7.700 | 100 | 48.1 | 7.650 | 100 |
| 25 | 40.2 | 7.882 | 95.89 | 43.9 | 8.032 | 96.87 | 42.3 | 7.619 | 93.56 | 38.5 | 7.815 | 95.94 | 43.5 | 7.699 | 93.88 |
| 50 | 38.6 | 7.758 | 87.06 | 40.3 | 8.011 | 86.36 | 39.5 | 7.598 | 88.38 | 36.5 | 7.801 | 89.93 | 39.5 | 7.581 | 87.16 |
| 75 | 36.2 | 7.634 | 82.59 | 37.8 | 7.914 | 84.67 | 34.6 | 7.455 | 81.77 | 33.3 | 7.691 | 86.07 | 36.6 | 7.538 | 84.58 |
| 100 | 30.1 | 7.544 | 77.65 | 32.9 | 7.873 | 80.59 | 29.1 | 7.363 | 78.34 | 29.6 | 7.653 | 81.95 | 31.2 | 7.484 | 82.85 |
| 125 | 26.3 | 7.519 | 74.41 | 29.0 | 7.844 | 78.34 | 26.8 | 7.279 | 75.03 | 27.9 | 7.610 | 79.85 | 28.3 | 7.445 | 76.49 |
| 150 | 23.6 | 7.425 | 70.53 | 26.9 | 7.767 | 72.69 | 23.5 | 7.202 | 71.28 | 25.3 | 7.559 | 73.52 | 25.3 | 7.411 | 73.94 |
| | UHTCC 12-20-2 | | | UHTCC 12-30-1 | | | UHTCC 12-30-2 | | | UHTCC 12-48-1 | | | UHTCC 12-48-2 | | |
| 0 | 52.3 | 7.600 | 100 | 48.5 | 7.600 | 100 | 54.5 | 7.605 | 100 | 50.6 | 7.600 | 100 | 55.3 | 7.850 | 100 |
| 25 | 47.9 | 7.806 | 94.11 | 41.8 | 7.726 | 95.62 | 49.3 | 7.650 | 98.81 | 44.7 | 7.670 | 94.08 | 50.9 | 7.972 | 97.36 |
| 50 | 43.8 | 7.744 | 86.52 | 39.1 | 7.658 | 88.21 | 46.3 | 7.648 | 95.21 | 42.1 | 7.537 | 90.02 | 47.5 | 7.994 | 93.35 |
| 75 | 79.6 | 7.586 | 83.92 | 35.9 | 7.555 | 83.92 | 41.9 | 7.596 | 89.26 | 38.2 | 7.495 | 85.42 | 42.9 | 7.811 | 90.65 |
| 100 | 35.7 | 7.536 | 81.36 | 32.3 | 7.495 | 80.16 | 37.6 | 7.564 | 84.60 | 33.3 | 7.442 | 82.57 | 38.9 | 7.774 | 86.71 |
| 125 | 33.6 | 7.499 | 77.14 | 28.9 | 7.401 | 78.32 | 34.8 | 7.510 | 79.44 | 31.2 | 7.371 | 78.86 | 36.3 | 7.721 | 84.25 |
| 150 | 29.6 | 7.451 | 75.95 | 26.5 | 7.337 | 72.69 | 32.3 | 7.458 | 76.74 | 28.9 | 7.305 | 75.96 | 34.5 | 7.681 | 80.76 |

### 3.1.2. Freeze-Thaw Mass Damage Model

The mass loss rate is an important parameter reflecting the structural damage inside the UHTCC and indicates the development process for accumulated damage inside the structure. The UHTCC freeze–thaw mass damage model is shown in Figure 5. The fitted coefficient of determination shows a wide fluctuation and does not have a high fitting accuracy. As the number of freeze–thaw cycles increases, the mass loss of the UHTCC specimens is the first to decrease and then increase. The reduction in mass loss is due to saturation of the concrete specimen with water absorption via the opening up of cracks inside the specimen at the beginning of the freeze–thaw cycle test. The latter increase is due to the shedding of cementitious material from the surface of the specimen with the aggregates. When the number of freeze–thaw cycles is less than 100, the accumulated mass loss rate for UHTCC specimens is relatively flat and basically controlled within 2.5%, showing a better frost resistance. When the number of freeze–thaw reaches 150, the accumulated mass loss rates for the UHTCC 12-30-2 and UHTCC 12-20-2 specimens are optimally 1.86% and 1.96%, respectively. The maximum accumulated mass loss rates for the UHTCC 5-30-1 and UHTCC 8-30-1 specimens are 4.81% and 5.24%, respectively. From the experimental data, it is concluded that the modified PP fibers with 12 mm length, 30 μm fiber diameter and 2% fiber doping show better frost resistance.

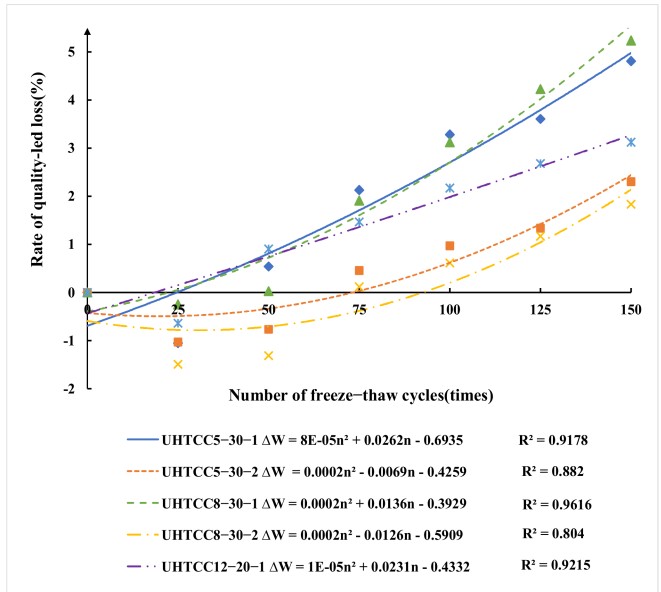
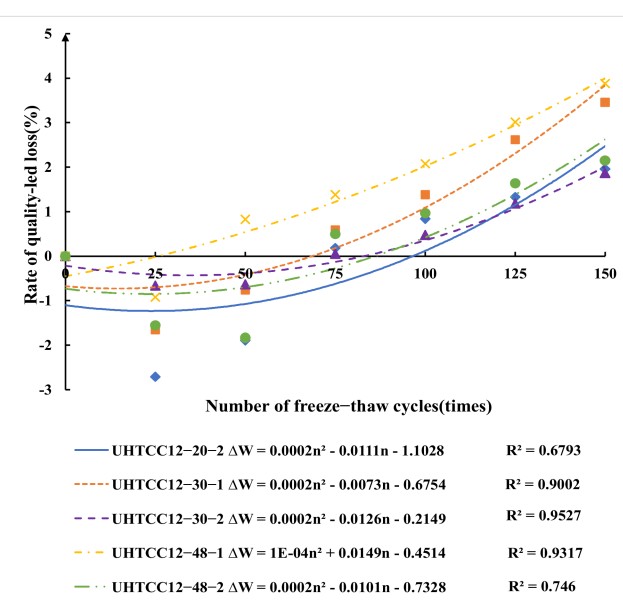

**Figure 5.** Mass damage under different number of freeze–thaw cycles.

### 3.1.3. Freeze–Thaw Relative Dynamic Elastic Model Damage Model

UHTCC damage in freeze–thaw environments results from the repeated accumulation of fatigue. When the pore water inside the specimen freezes and melts, a situation similar to a cyclic load acting on the pore structure inside the specimen occurs. In addition, when the temperature decreases, the frost swelling pressure increases, and the frost swelling pressure of the porous structure of UHTCC produces expansion products. Expansion products accelerate the generation and development of cracks and destroy the bond between the aggregate and cement. Thus, a penetration mechanism is formed, and the concrete becomes more brittle and suffers higher internal damage. The relative dynamic modulus of elasticity is the most common and effective nondestructive testing parameter used to evaluate the performance of concrete, which not only reflects the internal compactness of concrete but also yields the change in concrete under continuous damage. Figure 6 reflects the relative dynamic elastic modulus damage model for UHTCC. During the freeze–thaw cycle test, the relative dynamic modulus of elasticity for UHTCC undergoes two processes: steady

decline and accelerated decline. When the number of freeze–thaw cycles is less than 100, the relative dynamic modulus of elasticity of UHTCC can be controlled within 80%, showing good stability; when the number of freeze–thaw cycles reaches 150, the relative dynamic modulus of elasticity residuals for UHTCC 5-30-1 and UHTCC 8-30-2 specimens are at least 70.5% and 71.3%, and the relative dynamic modulus of elasticity residuals for UHTCC 12-48-2 specimens are at most 80.8%. The results show that the UHTCC specimens with a modified PP fiber length of 12 mm, fiber diameter of 48 μm and fiber doping of 2% show better frost resistance.

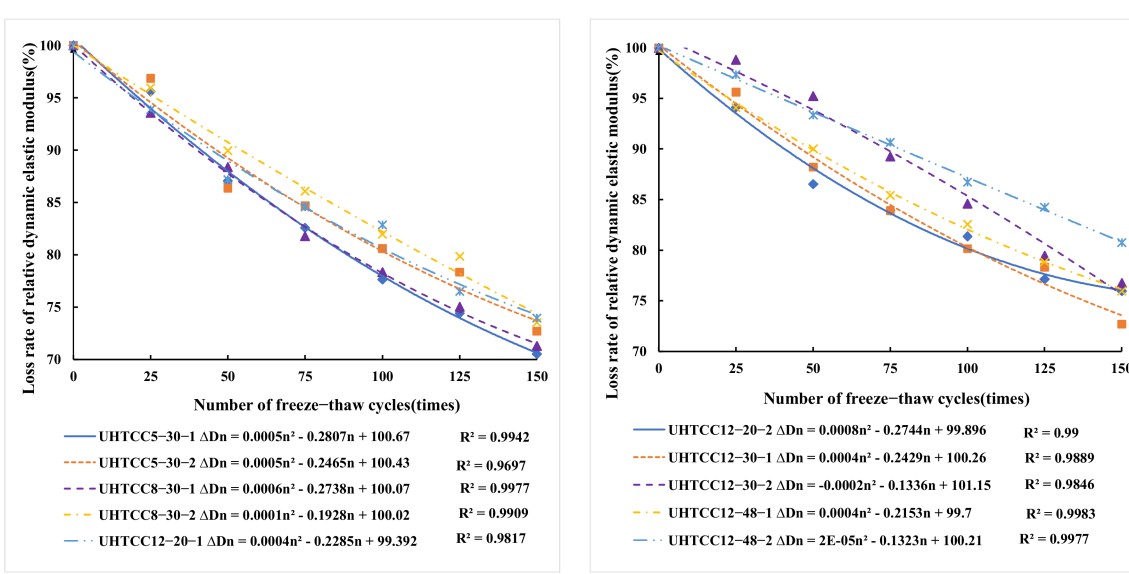

**Figure 6.** Relative dynamic elastic modulus damage under different number of freeze–thaw cycles.

### 3.2. Impact Resistance Test Results

The combined effect of different modified PP fiber lengths, fiber admixtures and fiber diameters on the impact resistance of PP-UHTCC specimens was studied. The number of impacts for the initial cracking and final cracking of the specimen under the action of the falling hammer impact is shown in Table 4. It can be observed from the data that the impact resistance of UHTCC is improved with the incorporation of modified PP fibers, which shows a certain improvement for the toughness of the concrete. The specimen shows the best impact resistance when the modified PP fiber length is 12 mm, the fiber diameter is 48 μm, and the fiber doping is 2%. According to Table 4, it can be observed that the impact number $N_1$ at initial cracking and the impact number $N_2$ at final cracking obtained during the impact resistance test shows a large dispersion. Therefore, the average values for the impact numbers for initial and final cracking of six base specimens were recorded in the test, and the impact energy dissipation W and ductility ratio $\beta$ were calculated according to Equations (1) and (2). The calculation results obtained are shown in Table 5, indicating that the addition of modified PP fibers better improves the impact resistance and ductility of UHTCC.

**Table 4.** The number of impacts at the initial cracking of the specimen is $N_1$, and the number of impacts at the final cracking is $N_2$.

| Codes | $N_1/N_2$ | | | | | |
|---|---|---|---|---|---|---|
| | **A** | **B** | **C** | **D** | **E** | **F** |
| UHTCC 5-30-1 | 1358/1365 | 1318/1325 | 1452/1466 | 1408/1416 | 1512/1524 | 1298/1311 |
| UHTCC 5-30-2 | 1578/1592 | 1624/1627 | 1502/1513 | 1678/1683 | 1549/1642 | 1689/1701 |
| UHTCC 8-30-1 | 1469/1473 | 1513/1314 | 1589/1602 | 1408/1416 | 1642/1658 | 1401/1402 |
| UHTCC 8-30-2 | 1711/1730 | 1625/1633 | 1824/1833 | 1746/1753 | 1825/1834 | 1658/1663 |
| UHTCC 12-20-1 | 1548/1566 | 1458/1463 | 1611/1635 | 1489/1503 | 1646/1653 | 1741/1752 |
| UHTCC 12-20-2 | 1892/1901 | 1805/1824 | 1943/1957 | 2015/2022 | 1743/1753 | 1845/1858 |
| UHTCC 12-30-1 | 1753/1758 | 1815/1823 | 1963/1968 | 1703/1719 | 1879/1892 | 1792/1799 |
| UHTCC 12-30-2 | 1683/1692 | 1547/1563 | 1746/1758 | 1625/1639 | 1845/1863 | 1941/1945 |
| UHTCC 12-48-1 | 1873/1886 | 1805/1809 | 1924/1936 | 2011/2013 | 1742/1747 | 1836/1839 |
| UHTCC 12-48-2 | 2109/2124 | 2236/2239 | 2338/2341 | 1953/1955 | 2313/2317 | 2163/2165 |

**Table 5.** Analysis of the impact resistance index of UHTCC specimens.

| Codes | Average Number of Cracking Impacts | | | $\beta$ | $W$ (J) |
|---|---|---|---|---|---|
| | $N_1$ | $N_2$ | $N_2 - N_1$ | | |
| UHTCC 5-30-1 | 1391 | 1401 | 10 | 0.7309 | 30,895.725 |
| UHTCC 5-30-2 | 1602 | 1610 | 8 | 0.5305 | 35,518.875 |
| UHTCC 8-30-1 | 1562 | 1572 | 10 | 0.6401 | 34,669.95 |
| UHTCC 8-30-2 | 1731 | 1740 | 9 | 0.5486 | 38,389.05 |
| UHTCC 12-20-1 | 1582 | 1595 | 13 | 0.8321 | 35,177.1 |
| UHTCC 12-20-2 | 1873 | 1885 | 12 | 0.6583 | 41,582.625 |
| UHTCC 12-30-1 | 1817 | 1826 | 9 | 0.4952 | 40,274.325 |
| UHTCC 12-30-2 | 1731 | 1743 | 12 | 0.7028 | 38,440.5 |
| UHTCC 12-48-1 | 1865 | 1871 | 6 | 0.3484 | 41,270.25 |
| UHTCC 12-48-2 | 2185 | 2190 | 8 | 0.2212 | 48,293.175 |

*3.3. Modified PP Fiber Contribution*

To study the effect of modified PP fibers on the compressive strength of UHTCC specimens after freeze–thaw cycling, the number of freeze–thaw cycles and fiber L/D ratio were used as independent variables, and the contribution of modified PP fibers was used as the dependent variable to define the fiber contribution of modified polypropylene fibers to quantify the degree of fiber influence. The larger the calculated value, the better the effect of the modified polypropylene fiber to mitigate the loss of the compressive strength of UHTCC specimens during freeze–thaw cycling and improve their frost resistance, as calculated in Equation (3).

$$Q_\omega = \frac{f_{\omega,n}}{f_{\omega,0}} \tag{3}$$

where: $Q_\omega$ denotes the contribution of polyvinyl alcohol fiber at volume content; $f_{\omega,n}$ denotes the compressive strength of UHTCC after n freeze–thaw cycling at volume content; $f_{\omega,0}$ denotes the compressive strength of unfreeze-thaw cycled UHTCC at $\omega$ content; $\omega$ denotes the volume content of modified polypropylene fiber. The relationship between the dose of modified PP fibers and their contribution after 0, 25, 50, 75, 100, 125, and 150 times freeze–thaw cycles was fitted, as shown in Figure 7.

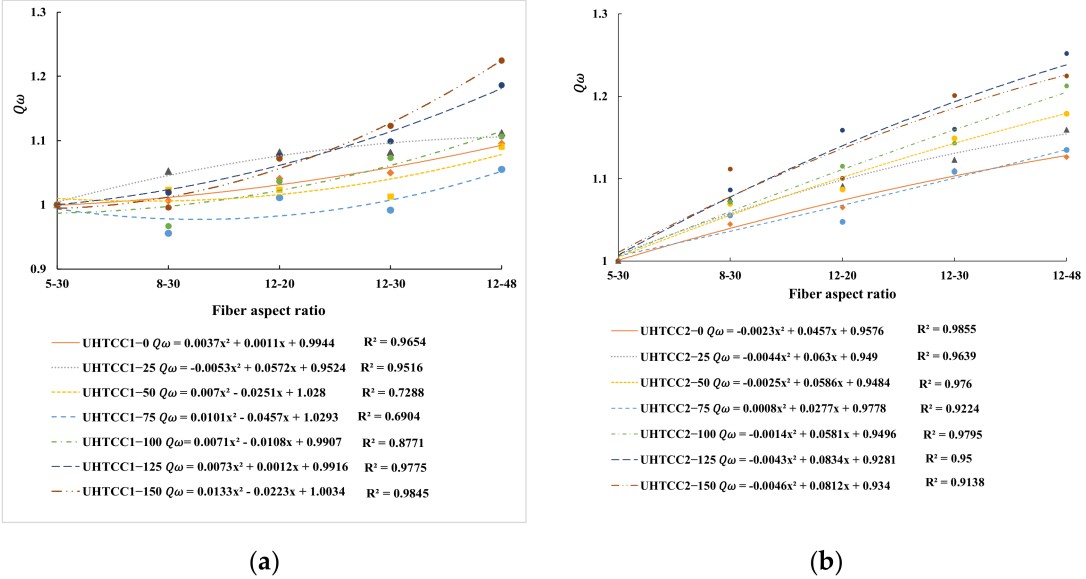

**Figure 7.** The relationship between the contribution rate and aspect ratio of modified PP fiber: (**a**) fiber content of 1%; (**b**) fiber content of 2%.

From the analysis of Figure 7, it can be observed that with the freeze–thaw cycle test, the contribution of modified PP fibers shows an increasing trend, and with increasing fiber length and diameter, the contribution of the fibers continues to increase. A fiber length of 12 mm, fiber diameter of 48 μm, and fiber content of 2% show the best improvement for the frost resistance of UHTCC, and the fitted coefficient of determination $R^2$ is generally greater than 0.9, and the fitting accuracy and significance are obvious. From a comparison of Figure 5 UHTCC with 2% fiber content shows better frost resistance, better resistance to the loss of compressive strength during freeze–thaw cycling, and a more stable fiber contribution.

## 4. Gray GM(1,1)-Based Freeze–Thaw Damage Model for UHTCC

The more applied gray system theory is the gray model (Gray Model), with the GM(1,1) power model abbreviated as gray. The brief principle is to first use the accumulation technique to generate data with an exponential law and then establish a first-order differential equation to solve it so that the result for the accumulation reduction can be used to obtain the gray prediction value. The gray GM(1,1) power model requires a data series with smooth changes in the data columns that are consistent with the exponential function characteristics. In this research, a combination of the variable-weight construction background value and residual correction optimization method was used to improve the prediction accuracy and smoothness and expand the application range. Based on the principle of minimizing the relative error of the average fit, the background values were constructed by selecting the best weights through automatic iteration theory to solve the systematic errors that exist when modeling. From the perspective of the quantitative relationship between the background values and the development coefficients, based on the least squares theory, an automatic optimization-seeking and weighting method is proposed to select the background values to optimize the prediction accuracy, establish the albinism differential equation, and solve the resulting time response equation.

### 4.1. Traditional GM(1,1) Power Model

The original nonnegative data sequence is $X^{(0)}$:$X^{(0)} = \left(X_1^{(0)}, X_2^{(0)}, \ldots, X_n^{(0)}\right)$ and the first-order cumulative calculation is conducted for $X^{(0)}$. A first-order cumulative sequence is generated: $1 - AGO$ as $X^{(1)}$, and $X^{(1)}$ can weaken the fluctuation of the $X^{(0)}$ data column.

$$X^{(1)} = \left(X_1^{(1)}, X_2^{(1)}, \ldots, X_n^{(1)}\right) \tag{4}$$

where: $X_k^{(1)} = \sum\limits_{i=1}^{k} X_1^{(0)} + X_2^{(0)} + \ldots + X_k^{(0)}, k = 1, 2, \ldots, n$.

The sequence $Z^{(1)}$ adjacent to the mean value can be obtained by piecewise summation of (5), and $Z^{(1)}$ is the background value of the GM(1,1) model.

$$Z^{(1)} = \left(Z_2^{(1)}, Z_3^{(1)}, \ldots, Z_n^{(1)}\right) \tag{5}$$

where: $Z_k^{(1)} = 0.5\left(X_{k-1}^{(1)} + X_k^{(1)}\right), k = 2, 3, \ldots, n$.

That is, the gray differential equation for constructing the model is expressed in Equation (6):

$$X_k^{(0)} + aZ_k^{(1)} = b \tag{6}$$

The first-order linear differential equation is the albinism equation of the gray differential equations:

$$\frac{dX_t^{(1)}}{dt} + aX_t^{(1)} = b \tag{7}$$

where parameter $a$ is the development coefficient, and parameter $b$ reflects the data variation relationship and is called the gray action. A parameter array is constructed: $\hat{\alpha} = [a, b]^T$.

$$\hat{\alpha} = [B^T B]^{-1} B^T Y \tag{8}$$

$$Y = \begin{bmatrix} X_2^{(0)} \\ X_3^{(0)} \\ \vdots \\ X_n^{(0)} \end{bmatrix}, \ B = \begin{bmatrix} -Z_2^{(1)} & \left(Z_2^{(1)}\right)^2 \\ -Z_3^{(1)} & \left(Z_3^{(1)}\right)^2 \\ \vdots & \vdots \\ -Z_n^{(1)} & \left(Z_n^{(1)}\right)^2 \end{bmatrix} \tag{9}$$

Under the initial condition $X_0^{(1)} = X_1^{(1)}$, the time response equation of the albinism differential equation is:

$$X_t^{(1)} = \left(X_1^{(0)} - \frac{b}{a}\right)e^{-a(t-1)} + \frac{b}{a} \tag{10}$$

Let $t = k$, which gives the time response function of the gray differential equation:

$$\widehat{X_k^{(1)}} = \left(X_1^{(0)} - \frac{b}{a}\right)e^{-a(k-1)} + \frac{b}{a}, \ k = 1, 2, 3, \ldots, \tag{11}$$

The predicted values can be reduced by cumulative reduction, and the relative dynamic elastic modulus damage prediction model based on the traditional GM(1,1) power model is derived:

$$\widehat{X_k^{(0)}} = \widehat{X_k^{(1)}} - \widehat{X_{k-1}^{(1)}} = \left(X_1^{(0)} - \frac{b}{a}\right)(1 - e^a)e^{-a(k-1)}, k = 2, 3, \ldots, n \tag{12}$$

### 4.2. Optimization of the Gray Background Values

From Equation (10), the accuracy of the GM(1,1) power model depends on the values of the parameters $a$ and $b$, The parameter series leads to the conclusion that the background value is an important indicator of the prediction accuracy for the GM(1,1) power model. Integrate Equation (7) over the interval $[k-1,k]$:

$$\int_{k-1}^{k} \frac{dX_t^{(1)}}{dt} + a \int_{k-1}^{k} X_t^{(1)} dt = b \tag{13}$$

where $\int_{k-1}^{k} \frac{dX_t^{(1)}}{dt} = X_k^{(1)} - X_{k-1}^{(1)} = X_k^{(0)}$. Comparing Equation (4) with Equation (11), we can see that $\int_{k-1}^{k} X_t^{(1)} dt = Z_k^{(1)}$. In fact, the background value should be the area of the curved trapezoid enclosed by $X_t^{(1)}$ in the interval $[k-1,k]$ and the time axis, The traditional GM(1,1) power model background value $Z_k^{(1)} = 0.5\left(X_{k-1}^{(1)} + X_k^{(1)}\right), k = 2,3,\ldots,n$ for the default background value before and after the 1-AGO is fixed with equal weights, and the fixed weight value is 0.5. The trapezoidal area approximation is used to replace $\int_{k-1}^{k} X_t^{(1)} dt$, thus affecting the prediction accuracy of the model.

The integral median theorem for $\int_{k-1}^{k} X_t^{(1)} dt = Z_k^{(1)}$ yields that there exists $\varphi \in [-1,1]$, yielding the formula for the optimal background value.

$$Z_k^{(1)} = (1 - \varphi)X_k^{(1)} + \varphi X_{k-1}^{(1)} \tag{14}$$

$$\varphi^\circ = \min_{\varphi}\left(\frac{1}{n-1}\right)\sum_{k=2}^{n} \Delta(k) \tag{15}$$

Set the background value coefficient calculation to the parameter with variable weights $[-1,1]$, take the value interval, and take the appropriate step size. The relationship between the variable-weight parameter $\varphi$ and the parameters $a$ and $b$ is used as a constraint to establish the background value optimization calculation model shown in Equation (14), and the average relative error between the actual value series $X_k^{(1)}$ and the model simulated value $\widehat{X_k^{(0)}}$ is used as the basis for selecting the best weight $\varphi^\circ$; thus, the optimal weights $\varphi^\circ$ are automatically selected. Substitution of $\varphi^\circ$ into the corresponding parameter series $\hat{\alpha} = [a,b]^T$ is used to build a gray prediction model, which in turn reduces the systematic error out of the prediction model. Optimization of the background values using this method does not change the structure of the GM(1,1) power model, and the corresponding gray differential equations remain consistent with the traditional GM(1,1) power model, which theoretically allows the model to achieve the highest prediction accuracy.

### 4.3. Freeze-Thaw Damage Model of UHTCC Based on Gray Prediction Theory

The background values for the test of the relative dynamic modulus of elasticity of UHTCC with the number of freeze–thaw cycles are shown in Table 6. From the data given in Table 6, the prediction model of the relative dynamic modulus of UHTCC under the influence of different fiber lengths, fiber diameters and fiber admixtures can be obtained according to Equations (4) and (15), and the model parameters are shown in Table 7.

**Table 6.** Background value of the UHTCC relative dynamic elastic modulus.

| Types | | \multicolumn{7}{c}{Freeze-Thaw Times} | | | | | | |
|---|---|---|---|---|---|---|---|---|
| | | 0 | 25 | 50 | 75 | 100 | 125 | 150 |
| UHTCC5-30-1 | Background value | - | 147.8 | 239.15 | 324 | 404.1 | 480.1 | 552.55 |
| | True value | 100 | 95.6 | 87.1 | 82.6 | 77.6 | 74.4 | 70.5 |
| UHTCC5-30-2 | Background value | - | 148.4 | 240 | 325.5 | 408.1 | 487.55 | 563.05 |
| | True value | 100 | 96.8 | 86.4 | 84.6 | 80.6 | 78.3 | 72.7 |
| UHTCC8-30-1 | Background value | - | 146.75 | 237.7 | 322.8 | 402.85 | 479.5 | 552.65 |
| | True value | 100 | 93.5 | 88.4 | 81.8 | 78.3 | 75 | 71.3 |
| UHTCC8-30-2 | Background value | - | 147.95 | 240.85 | 328.85 | 412.85 | 493.75 | 570.45 |
| | True value | 100 | 95.9 | 89.9 | 86.1 | 81.9 | 79.9 | 73.5 |
| UHTCC12-20-1 | Background value | - | 146.45 | 236.5 | 322.85 | 407 | 486.65 | 562.05 |
| | True value | 100 | 92.9 | 87.2 | 85.5 | 82.8 | 76.5 | 74.3 |
| UHTCC12-20-2 | Background value | - | 147.05 | 237.35 | 322.55 | 405.15 | 484.35 | 560.85 |
| | True value | 100 | 94.1 | 86.5 | 83.9 | 81.3 | 77.1 | 75.9 |
| UHTCC12-30-1 | Background value | - | 147.8 | 239.7 | 325.75 | 407.75 | 486.95 | 562.45 |
| | True value | 100 | 95.6 | 88.2 | 83.9 | 80.1 | 78.3 | 72.7 |
| UHTCC12-30-2 | Background value | - | 149.4 | 246.4 | 338.65 | 425.6 | 507.6 | 585.65 |
| | True value | 100 | 98.8 | 95.2 | 89.3 | 84.6 | 79.4 | 76.7 |
| UHTCC12-48-1 | Background value | - | 147.05 | 239.1 | 326.8 | 410.8 | 491.55 | 569 |
| | True value | 100 | 94.1 | 90 | 85.4 | 82.6 | 78.9 | 76 |
| UHTCC12-48-2 | Background value | - | 148.65 | 244 | 336 | 424.65 | 510.15 | 592.7 |
| | True value | 100 | 97.3 | 93.4 | 90.6 | 86.7 | 84.3 | 80.8 |

**Table 7.** Prediction model and coefficient of the relative dynamic elastic modulus of UHTCC.

| Types | Parameter $a$ and $b$ | Relative Dynamic Elastic Modulus Prediction Model |
|---|---|---|
| UHTCC5-30-1 | $a = 0.0599, b = 102.7544$ | $\hat{X}_k^{(1)} = -1715.43e^{-0.0599k} + 1815.43$ |
| UHTCC5-30-2 | $a = 0.0514, b = 101.8730$ | $\hat{X}_k^{(1)} = -1981.96e^{-0.0514k} + 2081.96$ |
| UHTCC8-30-1 | $a = 0.0367, b = 91.56690$ | $\hat{X}_k^{(1)} = -2495.01e^{-0.0367k} + 2595.01$ |
| UHTCC8-30-2 | $a = 0.0495, b = 102.6324$ | $\hat{X}_k^{(1)} = -2073.38e^{-0.0495k} + 2173.38$ |
| UHTCC12-20-1 | $a = 0.0438, b = 98.98930$ | $\hat{X}_k^{(1)} = -2260.03e^{-0.0438k} + 2360.03$ |
| UHTCC12-20-2 | $a = 0.0423, b = 98.35130$ | $\hat{X}_k^{(1)} = -2325.09e^{-0.0423k} + 2425.09$ |
| UHTCC12-30-1 | $a = 0.0512, b = 101.6710$ | $\hat{X}_k^{(1)} = -1985.76e^{-0.0512k} + 2085.76$ |
| UHTCC12-30-2 | $a = 0.0533, b = 107.3410$ | $\hat{X}_k^{(1)} = -2013.90e^{-0.0533k} + 2113.90$ |
| UHTCC12-48-1 | $a = 0.0430, b = 100.1556$ | $\hat{X}_k^{(1)} = -2329.20e^{-0.0430k} + 2429.20$ |
| UHTCC12-48-2 | $a = 0.0366, b = 102.6111$ | $\hat{X}_k^{(1)} = -2803.58e^{-0.0366k} + 2903.58$ |

*4.4. Error Analysis and Accuracy Analysis*

By substituting $k = 25$, 50, 75, 100, 125, 150 into the relative dynamic elastic modulus prediction model in Table 5, and substituting the prediction result $\hat{X}_k^{(1)}$ into the first-order cumulative calculation model in the inverse derivation yields $\hat{X}_k^{(0)}$, it is possible to use the GM(1,1) power-based model to calculate the initial predicted values of the relative dynamic modulus of elasticity at different numbers of freeze-thaw cycles for different modified polypropylene fiber lengths, diameters and volume contents and the resulting relative errors; see Figure 8 for details.

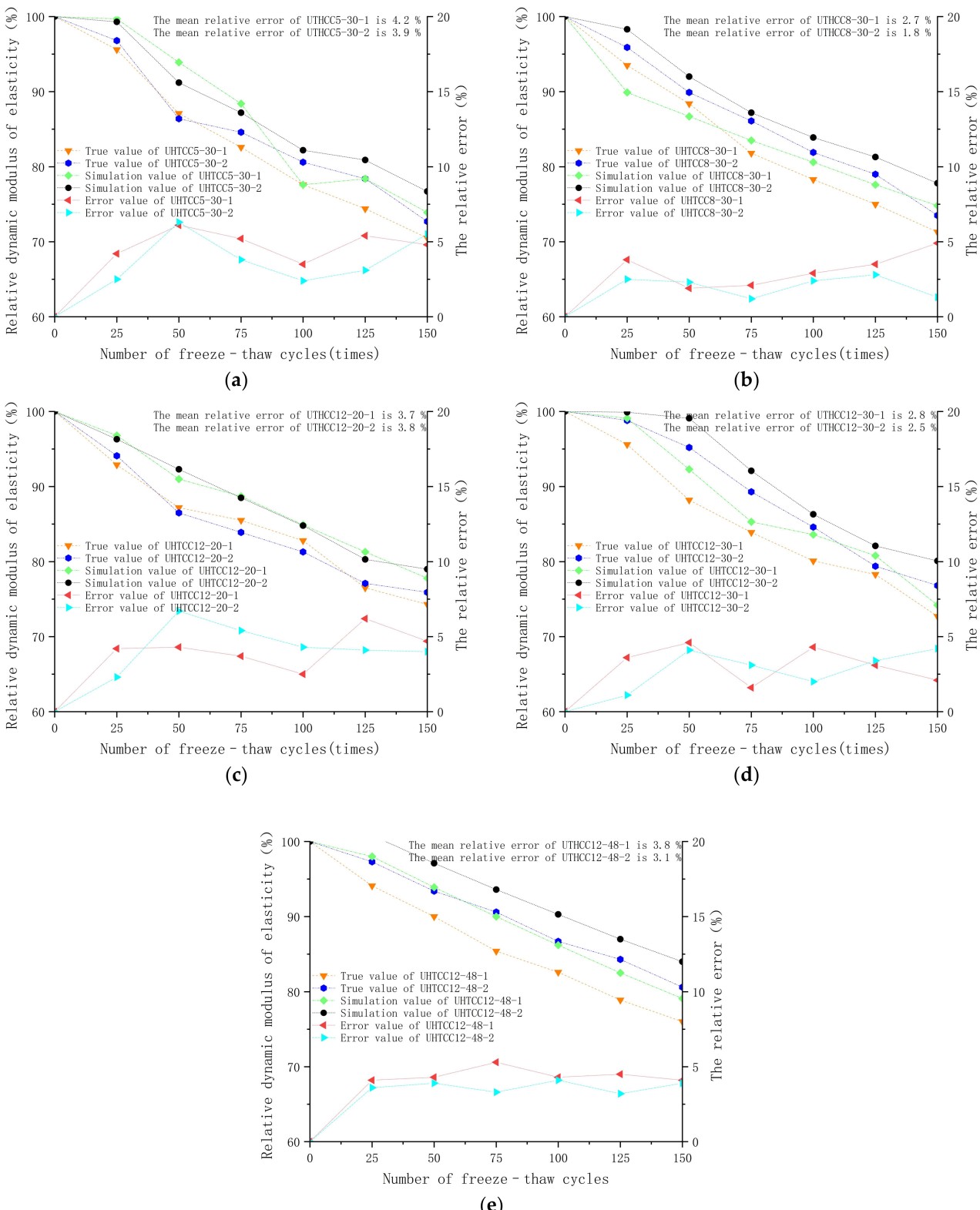

**Figure 8.** Error analysis of the GM(1,1) power model: (**a**) uhtcc5-30; (**b**) UHTCC8-30; (**c**) UHTCC12-20; (**d**) UHTCC12-30; (**e**) UHTCC12-48.

As shown in Figure 8, the relative errors between the predicted and true values of the UHTCC relative dynamic elastic modulus damage model based on the GM(1,1) power model are all relatively small, and the relative error value tends to change steadily with

increasing number of freeze–thaw cycles, indicating that the model has a high prediction accuracy. The maximum average relative error is 4.2% and the minimum average relative error is 1.8%, which are both below 5%, and more than half of the models show an average relative error below 3%, as can be derived from the figure. The peaks and valleys of the mean relative error for the GM(1,1) relative dynamic elastic modulus freeze–thaw damage model are within the controllable range with the variation in the modified PP fiber length, diameter and volume content, and no large fluctuations are observed. In summary, the UHTCC freeze–thaw damage model based on the GM(1,1) power model has a higher reliability and can be applied to the prediction of UHTCC freeze–thaw full-cycle damage.

To better analyze the applicability of the GM(1,1) power model to the freeze–thaw damage model of the relative dynamic elastic modulus of UHTCC, the model accuracy was determined by the combination of the relative error $\alpha$, mean variance ratio $C$, absolute correlation degree $\varepsilon$ and small probability error $P$. The absolute correlation and relative error were calculated to determine whether the error fluctuation is stable, and the mean variance and small probability error were calculated to determine the true reliability of the calculated error. For elements between systems, the magnitude of the correlation that leads to a change in the object over time is called the degree of correlation, and if the trend for the change of the factors is consistent, i.e., it shows a high degree of synchronous change, and the absolute correlation between the two can be said to be high. The gray absolute correlation analysis method distinguishes the degree of similarity or dissimilarity between factors according to the development trend as an important parameter to measure the degree of correlation between factors. The prediction levels for the model accuracy are shown in Table 8.

**Table 8.** Model established index critical value.

| Precision Determination Index | Model Accuracy | | | |
| --- | --- | --- | --- | --- |
| | Primary Standard | Secondary Standard | Three-Level Standard | Four-Level Standard |
| $C$ | 0.35 | 0.50 | 0.65 | 0.80 |
| $P$ | 0.95 | 0.80 | 0.70 | 0.60 |
| $\varepsilon$ | 0.90 | 0.80 | 0.70 | 0.60 |
| $\alpha$ | 0.01 | 0.05 | 0.10 | 0.20 |

Relative error $\alpha$:

$$\alpha = \left| \frac{X_k^{(1)} - \widehat{X_k^{(1)}}}{X_k^{(1)}} \right| \times 100\% \tag{16}$$

Absolute correlation degree $\varepsilon$:

$$\varepsilon_{ij} = \frac{1 + |s_i| + |s_j|}{1 + |s_i| + |s_j| + |s_i - s_j|} \tag{17}$$

$$|s_i| = \left| \sum_{k=2}^{n-1} X_k^{(0)} + \frac{1}{2} X_n^{(0)} \right| \tag{18}$$

$$|s_j| = \left| \sum_{k=2}^{n-1} X_k^{(1)} + \frac{1}{2} X_n^{(1)} \right| \tag{19}$$

$$|s_i - s_j| = \left| \sum_{k=2}^{n-1} \left( X_k^{(0)} - X_k^{(1)} \right) + \frac{1}{2} \left( X_n^{(0)} - X_n^{(1)} \right) \right| \tag{20}$$

Mean variance ratio $C$:

$$C = \frac{\mu_2}{\mu_1} \tag{21}$$

where $\mu_1$ is the mean variance of the original data and $\mu_2$ is the mean variance of the residuals:

$$\mu_1{}^2 = \frac{1}{n}\sum_{k=1}^{n}\left(X^{(0)} - \overline{X}\right)^2 \tag{22}$$

$$\mu_2{}^2 = \frac{1}{n}\sum_{k=1}^{n}\left(\partial(k) - \overline{\partial}\right)^2 \tag{23}$$

Probability of a small error $P$:

$$P = P\left\{\left|\partial(k) - \overline{\partial}\right| < 0.6745\mu_1\right\} \tag{24}$$

where $\partial(k)$ is the residual of the original series $X^{(0)}$ and the prediction model $\widehat{X_k^{(1)}}$, $\partial(k) = X^{(0)} - \widehat{X_k^{(1)}}$ and the mean of the residual is $\overline{\partial}$.

According to Equations (16)–(24), the relative error, absolute correlation degree, mean variance ratio and probability of the small error calculated for the UHTCC freeze–thaw damage model can be obtained as shown in Table 9. Under the combined effect of different modified PP lengths, diameters and volume contents, the small probability error and the mean variance ratio of the UHTCC freeze–thaw damage prediction model are in the first gradient class, indicating that the computational error of the model has high real reliability and meets the prediction accuracy required for the GM(1,1) power model. Previously, it has been shown [35] that relative dynamic elastic modulus damage prediction using GM(1,1) power model participation can be used for long-term prediction with very high prediction accuracy when the control system developmental dynamics parameter a is less than 0.3. From Table 9, the control system developmental dynamics parameters fitted using the GM(1,1) power model are all much less than 0.3, indicating that the GM(1,1) power model is suitable for full-cycle damage prediction for UHTCC.

**Table 9.** GM(1,1) power model error coefficient.

| Critical Value of the Index | The Relative Dynamic Elastic Modulus Prediction Accuracy of GM(1,1) Model | | | | | | | | | |
|---|---|---|---|---|---|---|---|---|---|---|
| | UHTCC 5-30-1 | UHTCC 5-30-2 | UHTCC 8-30-1 | UHTCC 8-30-2 | UHTCC 12-20-1 | UHTCC 12-20-2 | UHTCC 12-30-1 | UHTCC 12-30-2 | UHTCC 12-48-1 | UHTCC 12-48-2 |
| $C$ | 0.27 | 0.32 | 0.23 | 0.15 | 0.24 | 0.27 | 0.22 | 0.21 | 0.22 | 0.23 |
| $P$ | 1.00 | 1.00 | 1.00 | 1.00 | 1.00 | 1.00 | 1.00 | 1.00 | 1.00 | 1.00 |
| $\varepsilon$ | 0.9672 | 0.9681 | 0.9731 | 0.9815 | 0.9693 | 0.9685 | 0.9724 | 0.9738 | 0.9705 | 0.9719 |
| $\alpha$ | 0.042 | 0.039 | 0.027 | 0.018 | 0.036 | 0.038 | 0.028 | 0.025 | 0.038 | 0.031 |

## 5. Prediction of the Impact Damage Resistance for UHTCC Based on the Weibull Distribution Model

### 5.1. Weibull Distribution Model

The Weibull distribution model was proposed for the study of the fatigue life of materials. The model has a high prediction accuracy whether it is applied to the reliability analysis of structures or the life prediction of structures and is now widely used in academic fields as a data analysis method. The UHTCC material itself has many uncertainties because the incorporation of modified PP fibers also greatly improves the internal structure of the concrete. However, the concrete still contains a large number of irregularly distributed cracks and voids, and in the process of the UHTCC impact test, the impact damage is not uniformly superimposed, which also has an impact on the accuracy of concrete impact damage prediction. The number of impacts at final cracking of UHTCC specimens was used as a study variable, applied to the Weibull distribution model to analyze its feasibility and to establish the Weibull impact damage resistance evolution equation for UHTCC to study the effect of incorporation of modified PP fibers on the impact damage resistance test for UHTCC.

The distribution function in the Weibull distribution model is mainly composed of three parameters: shape parameter $\beta$, scale parameter $\eta$ and position parameter $x_0$. The scale parameter $\eta$ controls the degree of scaling of the Weibull distribution function, and the shape parameter $\beta$ controls the shape of the Weibull distribution function, Different probability density curves are obtained for different Weibull shape parameters, as shown in Table 10, and the expression for the three-parameter Weibull distribution function is shown in Equation (25).

$$f(x) = \begin{cases} 0 (x < x_0) \\ \frac{\beta}{\eta}\left(\frac{x-x_0}{\eta}\right)^{\beta-1} \exp\left[-\left(\frac{x-x_0}{\eta}\right)^{\beta}\right] (x \geq x_0) \end{cases} \tag{25}$$

**Table 10.** Relationship between shape parameters and probability density curves.

| Weibull Shape Parameters $\beta$ | Probability Density Curve Shape |
|---|---|
| $\beta = 1$ | Index distribution |
| $\beta = 2$ | Rayleign distribution |
| $3 < \beta < 4$ | Normal distribution |

*5.2. Analysis of UHTCC Impact Counts Based on the Weibull Distribution Model*

Since the impact damage mechanism for UHTCC shows a certain degree of similarity with the fatigue damage mechanism, the Weibull distribution model is introduced for probability density analysis of the impact resistance of the UHTCC specimen. The safety and reliability of UHTCC in service was considered. The coefficients in the Weibull distribution function are adjusted: so that the position parameter $x_0 = 0$; then, the original three-parameter Weibull distribution model is simplified to a two-parameter Weibull distribution function, as shown in Equation (26).

$$f(x) = \frac{\beta}{\eta}\left(\frac{x}{\eta}\right)^{\beta-1} \exp\left[-\left(\frac{x}{\eta}\right)^{\beta}\right] \tag{26}$$

The corresponding probability density distribution function can be obtained by integrating Equation (26), as shown by Equation (27).

$$f(x) = 1 - \exp\left[-\left(\frac{x}{\eta}\right)^{\beta}\right] \tag{27}$$

Next, the cumulative failure probability density function of UHTCC is given by Equation (28).

$$P_1(n) = 1 - \exp\left[-\left(\frac{n}{\eta}\right)^{\beta}\right] \tag{28}$$

The UHTCC survival function can be derived from Equation (28), as shown by Equation (29).

$$P_2(n) = 1 - P_1(n) = \exp\left[-\left(\frac{n}{\eta}\right)^{\beta}\right] \tag{29}$$

The relationship between the number of impacts and the probability of failure of UHTCC specimens was established by the Weibull distribution model, and the probability of failure of UHTCC specimens increases with increasing number of impacts. When the specimen reaches the ultimate number of impacts, $P_1(n) = 1$, both UHTCC specimens are damaged. Performing two natural logarithmic operations on both sides of Equation (29) yields Equation (30):

$$\ln\left[\ln\left(\frac{1}{p_2}\right)\right] = \beta ln\frac{1}{\eta} + \beta \ln(n) \tag{30}$$

Let $Z = \ln\left[\ln\left(\frac{1}{p_2}\right)\right]$, $Y = \ln(n)$; next, the above formula can be modified to Equation (31).

$$Y = a + bX \tag{31}$$

The approximate linear relationship between $Y$ and $X$ is expressed by Equation (31), and the linear fitted regression analysis leads to the parameters $a$ and $b$ and the coefficient of determination $R^2$. After linear fit regression analysis of the test data, the large $R^2$ value indicates that the Weibull distribution model can provide a reasonable analysis of the impact resistance for UHTCC. Under small sample conditions, the test values for the specimens were ranked from smallest to largest, and the average rank method was used to calculate the impact survival probability of UHTCC specimens, which leads to Equation (32):

$$P = 1 - \frac{c}{u+1} \tag{32}$$

where $u$ is the total number of samples per group of specimens and $c$ is the rank of the test data sorted from smallest to largest. Based on Equations (26)–(32), the Weibull parameters for the UHTCC can be calculated for each group of specimens, and a linear regression analysis with X as the horizontal coordinate and Y as the vertical coordinate was performed to obtain the linear regression curve for UHTCC, as shown in Figure 9. The corresponding regression parameters are shown in Table 11. Table 11 shows that the correlation coefficient of determination $R^2$ for the regression curve has a minimum value of 0.9393 and a maximum value of 0.9818, and the variation range of $R^2$ is relatively stable and generally at a high level. The results indicate that the simulation of the impact resistance times for UHTCC based on the two-parameter Weibull distribution model has a high fitting accuracy.

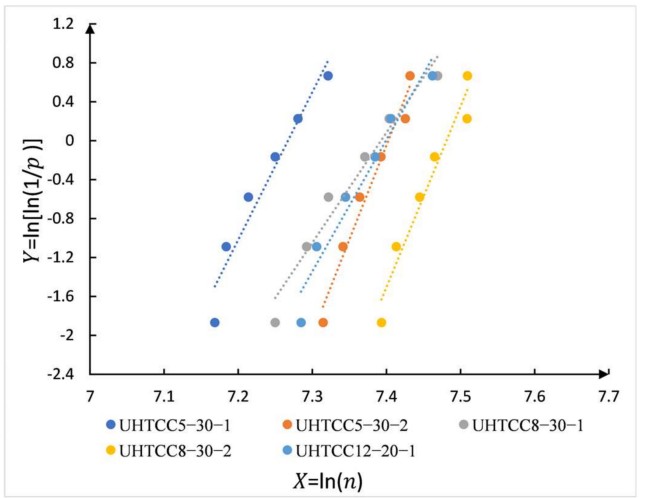
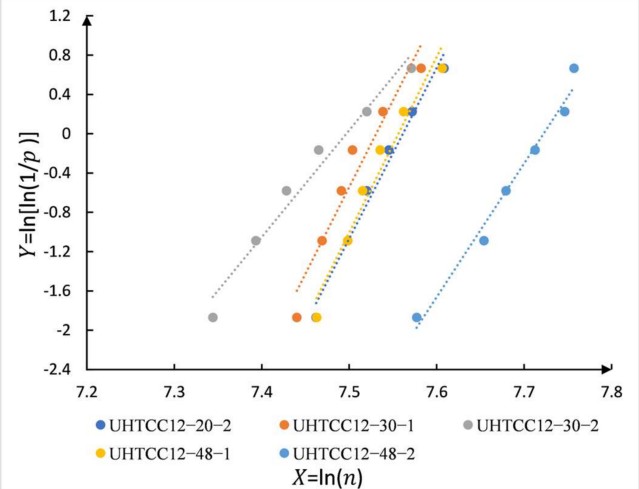

**Figure 9.** UHTCC Weibull Model Linear Regression Curve.

**Table 11.** PP-UHTCC Weibull regression parameters.

| Related Parameters | Specimen Number | | | | |
|---|---|---|---|---|---|
| | UHTCC 5-30-1 | UHTCC 5-30-2 | UHTCC 8-30-1 | UHTCC 8-30-2 | UHTCC 12-20-1 |
| $a$ | 15.18 | 19.378 | 11.316 | 18.591 | 13.542 |
| $b$ | $-110.32$ | $-143.44$ | $-83.659$ | $-139.08$ | $-100.2$ |
| $R^2$ | 0.9393 | 0.9713 | 0.9592 | 0.9491 | 0.9498 |
| | UHTCC 12-20-2 | UHTCC 12-30-1 | UHTCC 12-30-2 | UHTCC 12-48-1 | UHTCC 12-48-2 |
| $a$ | 17.381 | 17.633 | 10.878 | 17.893 | 13.621 |
| $b$ | $-131.42$ | $-132.79$ | $-81.548$ | $-135.21$ | $-105.19$ |
| $R^2$ | 0.9818 | 0.9408 | 0.9705 | 0.9582 | 0.9799 |

*5.3. Impact Damage Prediction Based on the Weibull Distribution Model*

The two-parameter Weibull distribution model was used to establish the UHTCC impact damage model, which was used to study the damage pattern for each group of specimens during the impact test. The Weibull distribution model failure probability function continuously increases with increasing number of impacts, and the probability of UHTCC impact damage increases with the monotonic increase of the resulting probability failure function. During the UHTCC impact test, it can be seen that the degree of fatigue damage to the specimen is considered as a superposition of each impact load, while the fatigue damage to the concrete is considered as a long-term accumulation process. The probability of damage and failure of UHTCC increases simultaneously during the impact test for the specimens. UHTCC has a failure probability of $P_1(n)$ and a damage degree of $D(n)$ after $n$ impact tests; when UHTCC reaches the limit impact number $N$ damage failure, the failure probability is $P_1(N) = 1$, and the damage degree is $D(N) = 1$. At this time $P_1(N) = D(N)$, both the failure probability and the damage degree will change simultaneously. In summary, the UHTCC two-parameter Weibull distribution model for impact damage prediction can be derived using Equation (33):

$$D(n) = 1 - \exp\left[-\left(\frac{n}{\eta}\right)^{\beta}\right] \tag{33}$$

The Weibull shape parameter $\beta$ and scale parameter $\eta$ in the Weibull distribution model of UHTCC can be found from Table 11 and Equation (31), and the shape parameter $\beta$ and scale parameter $\eta$ can be substituted into Equation (33) to derive the impact damage prediction model for the UHTCC Weibull distribution model, as shown in Table 12.

**Table 12.** UHTCC Weibull distribution model for impact damage prediction.

| Number | Impact Damage Prediction Model | Number | Impact Damage Prediction Model |
|---|---|---|---|
| UHTCC5-30-1 | $D(n) = 1 - \exp\left[-\left(\frac{n}{1432.902}\right)^{15.18}\right]$ | UHTCC12-20-2 | $D(n) = 1 - \exp\left[-\left(\frac{n}{1922.016}\right)^{17.381}\right]$ |
| UHTCC5-30-2 | $D(n) = 1 - \exp\left[-\left(\frac{n}{1639.602}\right)^{19.378}\right]$ | UHTCC12-30-1 | $D(n) = 1 - \exp\left[-\left(\frac{n}{1864.534}\right)^{17.633}\right]$ |
| UHTCC8-30-1 | $D(n) = 1 - \exp\left[-\left(\frac{n}{1624.546}\right)^{11.316}\right]$ | UHTCC12-30-2 | $D(n) = 1 - \exp\left[-\left(\frac{n}{1801.903}\right)^{10.878}\right]$ |
| UHTCC8-30-2 | $D(n) = 1 - \exp\left[-\left(\frac{n}{1774.083}\right)^{18.591}\right]$ | UHTCC12-48-1 | $D(n) = 1 - \exp\left[-\left(\frac{n}{1913.303}\right)^{17.893}\right]$ |
| UHTCC12-20-1 | $D(n) = 1 - \exp\left[-\left(\frac{n}{1634.68}\right)^{13.542}\right]$ | UHTCC12-48-2 | $D(n) = 1 - \exp\left[-\left(\frac{n}{2258.902}\right)^{13.621}\right]$ |

The change in the damage variables with the Weibull impact damage resistance model in Table 12 can be derived from the UHTCC damage change curve, as shown in Figure 10. From the impact damage curve, it can be seen that the ultimate impact resistance number for UHTCC gradually increases with an increase in the modified PP fiber length, volume

content and diameter. When the modified PP fiber length is 12 mm, the fiber diameter is 48 μm, and the volume fiber content is 2%, UHTCC shows the maximum ultimate impact resistance number, this content for UHTCC shows the best impact resistance, the prediction model data have high similarity with the test data, and the UHTCC impact damage resistance model based on the Weibull distribution model has high fitting accuracy.

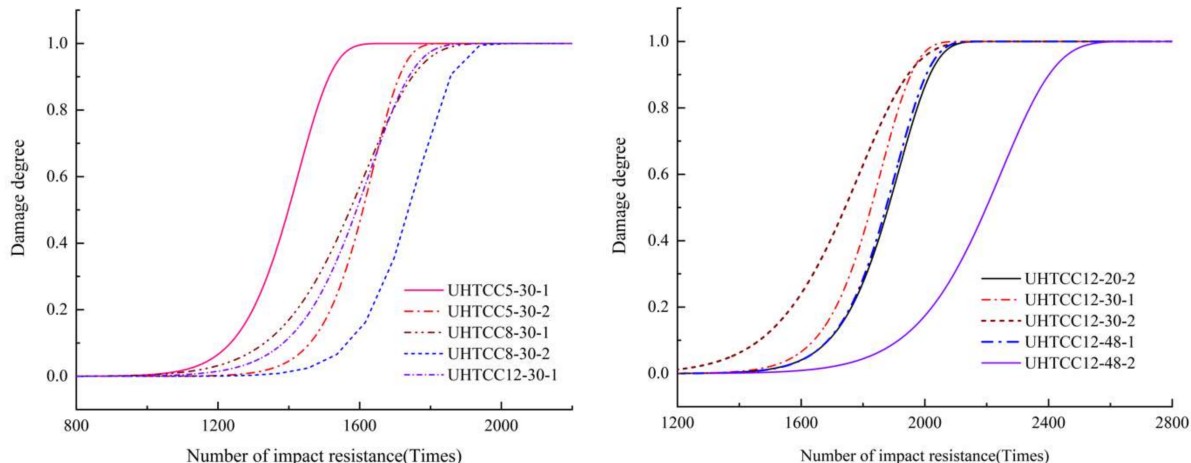

**Figure 10.** UHTCC impact damage curve.

## 6. Results and Discussion

(1) During the freeze–thaw cycle test, UHTCC shows better durable damage performance, and the frost resistance of UHTCC is greatly improved with the increasing of fiber length, fiber diameter and fiber volume content. During the impact test, the ultimate impact resistance number and ductility of UHTCC are greatly improved and showed excellent impact resistance toughness. The test data show that the UHTCC has excellent durability performance when the fiber length is 12 mm, the fiber diameter is 48 μm, and the fiber volume content is 2%.

(2) A freeze–thaw damage model based on the GM(1,1) power model with the UHTCC relative dynamic elastic modulus damage volume is developed. The accuracy and reliability of the GM(1,1) power model is analyzed based on the relative error, absolute correlation degree, mean variance and probability of small errors. Based on the experimental and predicted values, it can be concluded that the average relative error of the model is less than 5%, and the probability of a small error is 1. The absolute correlation and the mean variance are in the first gradient level, indicating that the UHTCC freeze–thaw damage model based on the GM(1,1) power model has a high fitting accuracy.

(3) An impact damage prediction model based on the Weibull distribution model and UHTCC final crack count is established and the impact damage curve is obtained. As the number of impacts resisted increases, the probability of failure of the UHTCC increases. The minimum value of the correlation coefficient of determination $R^2$ for the regression curve is 0.9393, the maximum value is 0.9818, and the variation range for $R^2$ is relatively stable, which indicates that the impact damage prediction model established for UHTCC based on the Weibull distribution model is highly reliable and lays the foundation for the promotion of practical applications.

**Author Contributions:** Conceptualization, C.W. and P.F.; methodology, C.W., P.F. and Z.L.; software, C.W., P.F. and Z.L.; validation, C.W., P.F. and Z.L.; formal analysis, C.W. and P.F.; investigation, C.W. and P.F.; resources, C.W., P.F., Z.L. and Z.X.; data curation, C.W., P.F. and Z.L.; writing—original draft preparation, C.W. and P.F.; writing—review and editing, C.W., P.F., Z.L., Z.X. and Y.L.; visualization, C.W., P.F. and Z.L.; supervision, C.W., P.F. and J.J.; project administration, C.W., P.F., T.W. and Y.Z.; funding acquisition, C.W., P.F. and J.J. All authors have read and agreed to the published version of the manuscript.

**Funding:** This research received no external funding.

**Data Availability Statement:** The data presented in this study are available on request from the corresponding author.

**Conflicts of Interest:** The authors declare no conflict of interest.

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
