# Peer review of "Study of the Durability Damage of Ultrahigh Toughness Fiber Concrete Based on Grayscale Prediction and the Weibull Model"

_buildings, doi:10.3390/buildings12060746_

Round 1

Reviewer 1 Report

 The technical aspect of this paper is acceptable. However, it is poor in this form in terms of grammar, lack of punctuation, use of long sentences, use of researcher's first name in the main body of the article (a few cases). Moreover, writing of researchers' name in reference section is nonuniform and inappropriate. Some references started with first name and some with the last name. In addition, use of et al. for researchers in the reference section is not acceptable and need to mention their full names.

Reviewer 2 Report

The paper investigated the durability damage law of UHTCC under freeze-thaw environment and impact test. The paper has well written. Some comments of the reviewer are as follows:

1) Introduction part, the authors should emphasize new approaches and advantages of the research.

2) Page 4, lines 154-155, the authors listed materials for UHTCC without coarse aggregates but in Table 2 UHTCC included coarse aggregates. Authors should check and revise.

3) Authors should select only a consistent word: UHTCC or UHTCC concrete

4) Line 222-225, authors should revise sentences to clearly explain how can determine the final cracking stage. An image of this stage after testing can be provided.

5) Authors should carefully check typos and revise. There were several typos. Besides, in a list of words, “, and” must be used to replace “and”.

Reviewer 3 Report

The paper entitled "Study on Durability Damage of Ultra-high Toughness Fiber Concrete Based on Gray-Scale Prediction and Weibull Model" is an interesting thematic paper that has been extensively documented.
However, the paper contains many technical errors and it is not systematized so that the reader understands the content of the paragraph.
The information presented are not systematized. Forced abbreviations are used (d for days and also d for the total number of samples per group of specimens). The title for 5.1. Subsection is not defined. Also, some translations do not conform, for example (line 196) "different freeze-thaw times" instead of "different freeze-thaw cycles".
Extensive verification of English language and style is required.
A major revision is needed.

Round 2

Reviewer 3 Report

The revised manuscript submitted, with the additions made by the authors, can be published.